# Developing a Microsatellite Polymerase Chain Reaction System for Small Yellow Croaker (*Larimichthys polyactis*) and Its Application in Parentage Assignment

**DOI:** 10.3390/biology13090710

**Published:** 2024-09-11

**Authors:** Eun-Soo Noh, Eun-Ha Shin, Hee-Jeong Kong, Young-Ok Kim, Yong-Woon Ryu

**Affiliations:** 1Biotechnology Research Division, National Institute of Fisheries Science, Busan 46083, Republic of Korea; laperm@korea.kr (E.-S.N.); shineunha@gmail.com (E.-H.S.); heejkong@korea.kr (H.-J.K.); yobest12@korea.kr (Y.-O.K.); 2Subtropical Fisheries Research Institute, National Institute of Fisheries Science, Jeju 63610, Republic of Korea

**Keywords:** small yellow croaker, genome information, parentage assignment, microsatellite marker

## Abstract

**Simple Summary:**

The small yellow croaker, a fish of high economic value in East Asia, faces population decline due to overfishing and environmental issues. The development of effective breeding programs for this species is crucial, and accurate parentage assignment is essential for such programs. This study developed a microsatellite marker-based multiplex PCR system for precise parentage identification in the small yellow croaker. The researchers utilized the assembled reference genome to select highly polymorphic markers and optimized a multiplex PCR system for simultaneous amplification. The system was validated using controlled mating pairs and successfully applied to a group mating scenario, demonstrating high accuracy in assigning offspring to their parents. This system offers a valuable tool for pedigree management, selective breeding, and conservation efforts for the small yellow croaker, facilitating sustainable aquaculture practices and genetic improvement.

**Abstract:**

(1) Background: The small yellow croaker, an economically important fish in East Asia, has been subjected to population declines due to overfishing and environmental pressures. The development of effective breeding programs is considered crucial for the species, and accurate parentage assignment is deemed essential for such programs. (2) Methods: The assembled reference genome of the small yellow croaker was utilized to select highly polymorphic microsatellite markers. A multiplex PCR system was optimized for the simultaneous amplification of these markers. The system’s accuracy was validated using controlled mating pairs and subsequently applied to a group mating scenario. (3) Results: The developed multiplex PCR system demonstrated high accuracy in assigning offspring to their parents in both the controlled and group mating scenarios. (4) Conclusions: The system is presented as a valuable tool for pedigree management, selective breeding, and conservation efforts for the small yellow croaker, facilitating sustainable aquaculture practices and genetic improvement.

## 1. Introduction

The small yellow croaker, *Larimichthys polyactis*, is a species belonging to the Sciaenidae family and the *Larimichthys* genus [1]. It holds significant economic importance in Korea, China, and Japan, making it one of the most renowned fish species in the region [2]. However, since the 1980s, the population of the small yellow croaker has experienced a significant decline, primarily due to overfishing, seawater pollution, changes in ocean currents, and alterations in water quality [3,4]. Consequently, the commercial value of the small yellow croaker has substantially increased [4,5]. Despite efforts aimed at restoring the species’ population, overfishing has resulted in critical issues, including early sexual maturity, gender imbalance, and a decrease in average size within small yellow croaker populations [2,6]. Therefore, the urgent establishment of artificial breeding techniques is necessary to address these challenges and rehabilitate fishery resources.

By utilizing artificial breeding methods, populations can maintain their genetic diversity, thereby increasing their resilience to environmental changes, diseases, and other stressors that may arise during breeding [7,8]. Moreover, genetic improvement programs require comprehensive genetic information to design effective breeding strategies aimed at acquiring economically valuable traits, such as growth regulation and survival [9,10]. Previous studies regarding the small yellow croaker have predominantly focused on resource evaluation and food nutrition [2,4,11].

Parentage assignment is one of the valid techniques for establishing legitimate pedigree relationships [12]. In aquaculture, the establishment of accurate pedigree relationships is critical, as it supports effective breeding programs and ensures the integrity of genetic lineages [13].

According to conventional practices, pedigree verification techniques involve the physical labeling of individuals in livestock and plant species using physical markers [14,15]. However, applying physical markers to most aquatic organisms, including the majority of fish species, presents significant challenges. The use of genetic markers in pedigree verification overcomes these limitations, facilitating lineage analysis even in aquatic organisms [16]. Microsatellites, also known as simple sequence repeats, are the most widely used molecular markers for pedigree analysis due to their codominance, multi-allelic nature, and high reproducibility, which result in high resolution [17]. Additionally, they exhibit high discriminatory power, making them extensively applied in population genetic research and forensic investigations [18].

In previous studies on the small yellow croaker, microsatellite markers were initially developed for 12 microsatellite loci, followed by various microsatellite loci based on EST [19,20]. More recently, with the advent of next-generation sequencing technologies, microsatellite loci based on genomic analysis have been developed [21]. These microsatellite markers have been utilized in population genetic studies. However, there are no current reports on the creation of a multiplex PCR system that utilizes microsatellite markers for parentage identification in the small yellow croaker [22,23]. Therefore, this present study aimed to develop a set of novel microsatellite markers and validate their effectiveness using artificially bred small yellow croakers to establish a reliable and efficient parentage assignment system.

## 2. Materials and Methods

### 2.1. Sample Collection and DNA Extraction

To isolate the high-molecular-weight genomic DNA from the muscle tissue of the small yellow croaker, 600 µL of 8M TNES-UREA and 5 µL of proteinase K were used. The mixture was incubated at 37 °C for 12 h. After confirming complete tissue lysis, the genomic DNA was purified using the phenol–chloroform extraction method and ethanol precipitation [24]. The separated genomic DNA was then analyzed for its presence using 1% agarose gel electrophoresis, and the concentration was measured using the NanoVue spectrophotometer (GeHealthcare, Fairfield, NJ, USA). For genetic species identification from the extracted small yellow croaker genomic DNA, the Cytochrome-c oxidase subunit I (COI) gene sequence of the mitochondrial DNA was analyzed. Universal primers VF2 (5′-TCA ACC AAC CAC AAA GAC ATT GGC AC-3′) and FishR2 (5′-ACT TCA GGG TGA CCG AAG AAT CAG AA-3′) were used to amplify the DNA and were analyzed on an ABI 3730XL DNA analyzer (Applied Biosystems, Waltham, MA, USA) [25]. A reference population comprising three families of small yellow croaker, each produced by artificial insemination from a mating pair (one male and one female), was used for marker validation. The final paternity test verification was conducted using a reference population generated through group mating. All experimental animals used in this study were obtained from the Subtropical Fisheries Research Institute, National Institute of Fisheries Science, Republic of Korea (Table 1).

### 2.2. Genome Sequencing and Assembly

The mitochondrial COI gene sequence (669 bp) obtained from the small yellow croaker was analyzed using the BLAST algorithm of NCBI (http://www.ncbi.nlm.nih.gov, accessed on 20 January 2023) and showed 99.85% identity with *Larimichthys polyactis* (accession number EU266386.1). Next-generation sequencing (NGS) was employed for the assembly of the reference genome and the selection of microsatellite markers. To conduct long-read sequencing, PacBio CLR libraries with fragments exceeding 20 Kb were generated, followed by high-capacity sequencing of over 50 Gb using the PacBio Sequel II system. For short-read sequencing, paired-end sequencing with a capacity exceeding 30 Gb was performed using the Illumina NovaSeq system in our study. To ensure the precision of the reference genome’s complete sequence information, we performed sequencing data preprocessing by eliminating contaminants and low-quality reads. This process entailed the removal of sequencing adapter sequences and sequences with a quality score lower than 20. Furthermore, we employed cross-validation with a microbial genome database to eliminate any tainted data. Subsequently, we conducted an estimation of the reference genome’s overall genome size using k-mer analysis based on high-quality sequence reads. In order to complete the standard genome sequence, we initially performed de novo contig assembly using the FALCON assembler [26], and the initial draft contigs underwent meticulous refinement through error correction using preprocessed short-read sequences. Finally, we updated the genome to the chromosome level using HiRise scaffolding with Omni-C reads, ultimately constructing the standard genome sequence.

### 2.3. Microsatellite Selection and Primer Design

Krait v1.3.3 software was utilized for the screening of microsatellite markers in the small yellow croaker [27]. Candidate markers were discovered from the assembled draft genome data of the small yellow croaker, employing different repeat thresholds based on the motif length. The prediction criteria were set as follows: di-nucleotide motifs with a minimum of 9 repeats, tri-nucleotide motifs with a minimum of 6 repeats, tetra-nucleotide motifs with a minimum of 5 repeats, penta-nucleotide motifs with a minimum of 4 repeats, and lastly, hexa-nucleotide motifs with a minimum of 3 repeats. The nucleotide sequence served as the repeating pattern requirement, while the motif length was constrained within the range of 7–30 base pairs. Among the selected markers, priority was given to microsatellite markers located in the 3′-UTR (untranslated region) that exhibited a relatively high level of stability and genetic diversity [28]. For the design of PCR primers, upstream and downstream sequences of approximately 500 base pairs each were obtained. The selected markers were designed using the Primer3 engine included in the Krait software. To establish multiplex conditions, considerations were made to achieve final product sizes of 100, 200, and 300 base pairs.

### 2.4. Amplification and Validation of Microsatellite Markers

The candidate microsatellite markers were validated by performing primer testing using DNA samples from the small yellow croaker (NFRDI-FI-TS-0064219, -0065913, -0065915) stored in the Fisheries bio-resources storage of the National Institute of Fisheries Science (Busan, Republic of Korea). TaKaRa Ex-Taq DNA Polymerase (TAKARA Bio. Inc., Kusatsu, Japan) was used for PCR amplification. The reaction mixture, based on a 20 µL volume, consisted of 1 µL of gDNA, 2 µL of 10X Ex Taq buffer, 1.6 µL of 2.5 mM dNTP mixture, 1 µL of 10 pmol forward primer, 1 µL of 10 pmol reverse primer, and 0.1 µL of Ex Taq polymerase. PCR amplification was conducted using the ABI verity fast thermal cycler (Applied Biosystems, Waltham, MA, USA) with the following cycling conditions: initial denaturation at 94 °C for 5 min, followed by 35 cycles of amplification (30 s at 94 °C, 45 s at 56 °C, and 45 s at 72 °C), and a final extension at 72 °C for 10 min.

### 2.5. Establishing Multiplex PCR Conditions

To efficiently streamline the analysis approach, the establishment of multiplex PCR conditions was pursued. This involved assessing PCR amplification success and considering product sizes. A total of 9 microsatellite markers were ultimately selected, and these markers were chosen to compliment overlapping product sizes by utilizing three fluorescence dyes (FAM, HEX, TAMRA) (Figure 1). The establishment of multiplex PCR conditions utilized the same polymerase as in previous studies. The composition of the mixture consisted of 2 µL of gDNA, 2 µL of 10X Ex Taq buffer, 1.6 µL of 2.5 mM dNTP mixture, 0.05 to 1 µL each of 10 pmol forward and reverse primers, and 0.5 µL of Ex Taq polymerase. The quantities of each primer were adjusted for optimization based on the analysis results.

### 2.6. Statistical Analysis and Parentage Assignment

The allele sizes of the SSR markers were determined using the ABI 3730XL DNA analyzer (Applied Biosystems, Forster City, CA, USA). Genotyping results were generated using GeneMapper v5.0 software (Applied Biosystems). Allele frequencies were calculated, and a principal component analysis (PCA) was performed using GeneAlEx v6.502 to assess the genetic relationships among the samples [29]. The marker efficiency for parentage verification was tested using the Cervus 3.0.7 software based on genotype data [30].

## 3. Results

### 3.1. Genome Assembly

Through next-generation sequencing analysis, we conducted long-read sequencing using PacBio Sequel, resulting in a total of 135.1 Gb. Short-read sequencing was performed using the Illumina NovaSeq platform (Illumina, San Diego, CA, USA), yielding a total of 31.67 Gb. Additionally, sequencing with the Omni-C library was carried out on the NovaSeq platform, producing a total of 36.64 Gb of data for our study, which will be utilized in this paper. After preprocessing short-read (paired-end) sequencing data, a total of 237.6 M reads were obtained, amounting to 34.96 Gbp of clean reads, with approximately 2.19% of reads being removed during the preprocessing step. Based on the k-mer analysis, the genome size of the small yellow croaker was estimated at 663.98 Mb. Using FALCON assembler v 0.2.2 on PacBio long reads, we obtained an initial contig assembly consisting of 400 contigs with a total size of 700 Mb. Subsequently, through the use of second-generation Omni-C reads and HiRise scaffolding, we finalized the standard genome sequence, comprising 24 chromosomes and 177 unknown contigs. The completed reference genome’s standard sequence was confirmed to be slightly larger at 700.58 Mb, compared to the previously predicted genome size of 663.98 Mb (Table 2).

### 3.2. Microsatellite Selection

An analysis was conducted based on single-nucleotide repeats (SSRs) with a length of ≥12 bp using the genome assembly data of the small yellow croaker. In this analysis, a total of 460,863 perfect SSRs were identified within the 700.58 Mb genome sequences. Dinucleotide repeats were the most abundant (252,568, 54.8%), followed by mono-nucleotide (92,333, 20.03%), tri-nucleotide (55,625, 12.07%), tetra-nucleotide (45,914, 9.96%), penta-nucleotide (10,467, 2.27%), and hexa-nucleotide (3956, 0.86%) repeats. Considering the need for consistency and accuracy in generation genotype information for parentage testing, 96 SSRs evenly distributed throughout the genome were initially selected based on tri-nucleotide repeats. Primers of 100 bp, 200 bp, and 300 bp, capable of amplifying the selected SSRs, were designed, and PCR amplification was performed on three small yellow croaker individuals for validation. Of the 96 candidate SSR markers, 56 were amplified, and 32 SSR markers were selected as final candidates based on the observed polymorphisms. Subsequently, the selected markers were fluorescently labeled with FAM, and an analysis of genotype information was successfully performed on nine final SSR markers (Table 3).

### 3.3. Marker Validation

The feasibility of parentage verification was evaluated using the nine final SSR markers. Sixteen offspring were obtained from each of three 1:1 mating pairs of male and female small yellow croakers. In addition, sixteen wild individuals were included as a control group in the analysis. Genotyping was performed using the nine SSR markers, and parentage verification was conducted. Principal component analysis was performed using GenAlEx software v6.502 to analyze the samples, and the results confirmed that the samples were clearly clustered by family (Figure 2). Parentage assignment analysis was performed using Cervus 3.0.7 software including parental information. Analysis of allele frequencies per marker revealed that offspring from the three mating groups inherited a minimum of three and a maximum of four alleles from their parents. This observation aligns with Mendelian inheritance principles, indicating the transmission of genetic traits from parents to offspring without the occurrence of mutations. Furthermore, to validate the reliability of the analysis, the presence of null alleles was assessed. The frequency of null alleles was found to be relatively low, ranging from −0.055 to 0.088, confirming the validity of the markers used in this study. Polymorphism information content (PIC) values, which indicate the degree of polymorphism of genetic markers, were high (0.732–0.884) for most loci, suggesting a high level of polymorphism overall (Table 4).

The parentage analysis revealed that the most accurate estimation of parentage in the small yellow croaker was achieved when using the genetic information of both parents. At the 95% confidence level, 75% of the offspring were assigned, with the remaining 25% belonging to the wild population, resulting in an overall accuracy of 100%. In contrast, when using only the mother’s or father’s information, only 75% and 67% of offspring were assigned at the 95% confidence level, respectively. Nevertheless, considering the inclusion of the wild population, the high accuracy achieved even with single-parent information underscores the effectiveness of the utilized genetic markers in distinguishing individuals (Table 5).

### 3.4. Establishing Multiplex PCR System

To optimize the multiplex PCR conditions for simultaneous amplification of the nine selected markers, various primer concentrations and annealing temperatures were tested. The final optimized PCR mixture, with a total volume of 20 µL, contained the following primer concentrations: Lp1 and Lp2 at 0.1 µL each, Lp5 at 0.5 µL, Lp7 and Lp10 at 0.05 µL each, Lp8 and Lp9 at 0.4 µL each, Lp12 at 1 µL, and Lp14 at 0.3 µL. Due to the complexity of multiple primers in the reaction, a touchdown PCR protocol was implemented. The optimal amplification was achieved with the following annealing temperature profile: 5 cycles at 59 °C, 5 cycles at 58 °C, and 32 cycles at 57 °C. This approach yielded the clearest and most reliable genotyping results.

### 3.5. Parentage Assignment

The validated genetic markers and established multiplex PCR system were used to perform parentage assignment analysis on a significantly larger group of small yellow croakers produced through group mating compared to the 1:1 mating pairs used for marker development. A total of 90 parental individuals with unknown sex information and 230 offspring produced from these parents were analyzed. The increased number of individuals in this analysis was necessary to ensure accurate results, as group mating scenarios involve an uncertain number of contributing parents and the potential presence of siblings and half-siblings among the offspring. The “sexes unknown” option in Cervus 3.0.7 software was employed for the analysis. Based on an LOD score threshold of 95%, 221 offspring were successfully assigned to their parents. Among these assigned offspring, the number of mismatches per individual ranged from 0 to 2 across the nine markers. Eight offspring were additionally assigned to candidate parents based on an LOD score threshold of 80%, with 0 to 3 mismatches observed. The remaining one offspring showed one or two mismatches with candidate parents but could not be assigned to any parental pair based on genotype combinations. Out of the 90 potential parents, 28 were identified as contributing to the offspring generation, resulting in 24 distinct families, with some individuals contributing to multiple families while others did not contribute at all (Appendix A).

## 4. Discussion

The development of a multiplex PCR system for parentage assignment in the small yellow croaker (*Larimichthys polyactis*) utilizing nine microsatellite markers represents a substantial advancement in the aquaculture and conservation efforts for this economically important species. The high accuracy and efficiency of this system, demonstrated through the successful assignment of offspring to their respective parents in both controlled and group mating scenarios, provides a robust tool for pedigree management and selective breeding programs.

The development of microsatellite markers and multiplex PCR systems for parentage assignment has been successfully implemented in various commercially valuable aquaculture species, including the Pacific oyster (*Crassostrea gigas*) [31], the olive flounder (*Paralichthys olivaceus*) [32], and the black tiger shrimp (*Penaeus monodon*) [33]. The present study contributes to this growing body of knowledge by providing a robust and efficient parentage assignment tool specifically tailored to the small yellow croaker, further expanding the applicability of such systems in aquaculture management and conservation efforts.

The selection of microsatellite markers from the assembled reference genome ensured a comprehensive representation of the genetic diversity within the species. These markers, evenly distributed across the genome, exhibit high levels of polymorphism, making them ideal for parentage assignment and other genetic analyses. The utilization of a multiplex PCR system further enhances the efficiency and cost-effectiveness of the genotyping process, positioning it as a practical tool for large-scale applications in both aquaculture settings and studies of wild populations.

The ability to accurately assign parentage in the small yellow croaker opens new avenues for selective breeding programs aimed at enhancing economically valuable traits such as growth rate, disease resistance, and flesh quality. By identifying superior parents and their offspring, breeders can make informed decisions to improve the productivity and profitability of aquaculture operations. Additionally, this parentage assignment system can serve as a foundation for future studies aimed at understanding the genetic diversity and population structure of wild stocks. These insights could inform conservation and management strategies to preserve genetic resources and prevent inbreeding depression in the face of environmental pressures.

The few instances of unassigned offspring in the group mating scenario highlight the potential challenges in parentage assignment when dealing with complex mating systems and underscore the need for continued refinement of the system. These unassigned offspring could result from genotyping errors, mutations, or the presence of unsampled parents in the breeding population. Further investigation into these cases, including the use of additional markers or more sophisticated statistical methods, could help to improve the accuracy of the system and provide a more comprehensive understanding of the genetic relationships within both cultured and wild populations.

## 5. Conclusions

This study successfully developed and validated a multiplex PCR system utilizing nine microsatellite markers for parentage assignment in the small yellow croaker. The markers were selected from the assembled reference genome and validated using offspring from known 1:1 mating pairs. The established multiplex PCR system proved to be highly accurate and efficient, allowing for the simultaneous amplification of all nine markers and their reliable genotyping in both controlled and group mating scenarios, as evidenced by the high assignment success rate. This demonstrates its effectiveness in identifying parent–offspring relationships in both research and aquaculture settings specific to the small yellow croaker. The development of this robust and cost-effective parentage assignment system marks a significant advancement in aquaculture efforts for the small yellow croaker, streamlining selective breeding programs aimed at improving economically important traits and ultimately contributing to the sustainable management and genetic improvement of this valuable fishery resource. The use of microsatellite markers in this study successfully demonstrates their effectiveness for parentage assignment in the small yellow croaker. However, it is worth noting that the field of molecular markers is constantly evolving, with single-nucleotide polymorphisms (SNPs) gaining increasing popularity due to their abundance and potential for high-throughput genotyping [34]. Future research could explore the development of SNP-based parentage assignment systems for the small yellow croaker, potentially offering even higher accuracy and efficiency compared to microsatellite markers. The integration of both marker types could provide a powerful tool for comprehensive genetic analysis and management of this species.

## Figures and Tables

**Figure 1 biology-13-00710-f001:**
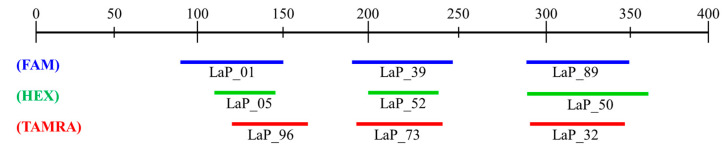
Size range of microsatellite markers for multiplex PCR.

**Figure 2 biology-13-00710-f002:**
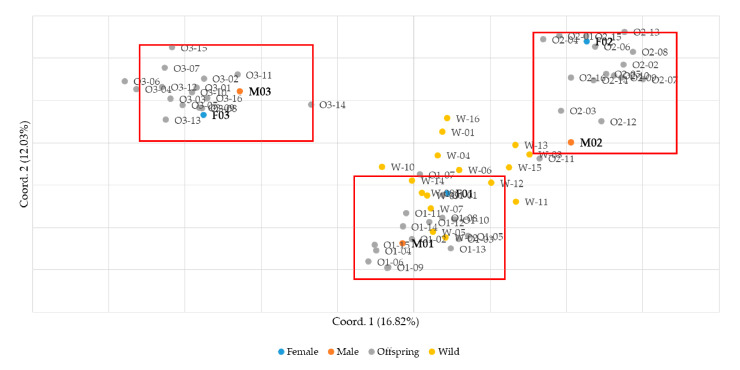
Analysis of genetic relationships between parents, offspring, and wild groups using PCA.

**Table 1 biology-13-00710-t001:** The sample information of *Larimichthys polyactis*.

Group	Breeding System	Generation	Sex	Number of Individuals
1	Artificial insemination	Parents	Male	3
			Female	3
		Offspring	-	48
2	Group mating	Parents	-	90
		Offspring	-	230
3	Wild type (Jeju Island)	-	-	16

**Table 2 biology-13-00710-t002:** Genome assembly metrics of *Larimichthys polyactis*.

Metrics	Contig Level	Chromosome Level
Number of Contigs	400	24 chromosomes + 177 unknowns
Residues	700,560,536	675,620,547 + 24,959,989
Ave. length	1,751,401.34	3,485,475.30
Min. length	31,123	31,123
Max. length	11,060,126	40,851,670
N50	4,969,043	28,972,644
N (%)	0 (0.00)	20,000 (0.00)
GC (%)	291,237,301 (41.58)	291,237,301 (41.58)

**Table 3 biology-13-00710-t003:** List of microsatellite markers for parentage analysis.

Name	Motif	Size Range	Sequence	Fluorescence
LaP_01	(AAG)_16_	82–148	(F) TCTGGTTGCAATTTACGCTGC	FAM
			(R) TCTCTCAAACTATCAGACACAAACC	
LaP_05	(TCT)_17_	110–143	(F) GACAAGGACAGGCTCAGTCG	HEX
			(R) AGACTGGCTCAGTGATCAGC	
LaP_32	(TAA)_24_	282–342	(F) CCTTTCTGAATTGGGCGTGG	TAMRA
			(R) CCACATCTCACTCTGTGAATTAACG	
LaP_39	(GTA)_24_	183–246	(F) CGTCTGCCATGTCAATGTGC	FAM
			(R) AAGGCTTACATCGGGTTGGG	
LaP_50	(ATT)_24_	278–362	(F) CCCCTGACATCTCATCCAGC	HEX
			(R) CTGTAGCTTAATTGCTTATTACTCTGC	
LaP_52	(AAT)_21_	200–230	(F) TTAGGAGACCTGTTTGCCGG	HEX
			(R) CACCAAGGCTACAGTTTGGC	
LaP_73	(GAA)_17_	190–235	(F) TCAGTGAAACCGAAGTGGGC	TAMRA
			(R) GTCCCTCTCGTTATTCTTCTTGC	
LaP_89	(ATA)_21_	276–348	(F) TTGTCAGACAGTAGGCACGG	FAM
			(R) ATTCAAGCATGCCACAGTCG	
LaP_96	(TTA)_15_	120–162	(F) TGCATATCATACGTCTCCCTCC	TAMRA
			(R) CAAACTTGGAACCCCATCCC	

**Table 4 biology-13-00710-t004:** Summary of allele frequency analysis results.

Locus		LaP_01	LaP_05	LaP_32	LaP_39	LaP_50	LaP_52	LaP_73	LaP_89	LaP_96
N_A_	Mating 1	4	3	4	3	3	3	4	4	4
	Mating 2	3	4	4	3	4	3	3	4	4
	Mating 3	4	2	3	3	4	4	2	4	3
	Wild	12	9	9	9	8	9	11	9	8
PIC		0.802	0.734	0.884	0.810	0.805	0.764	0.762	0.837	0.826
Null		−0.055	−0.039	0.088	0.075	−0.025	0.020	0.034	−0.053	0.028

N_A_—number of alleles, PIC—polymorphism information content.

**Table 5 biology-13-00710-t005:** Summary of parentage analysis results.

Analysis Type	Confidence Level (%)	Critical Delta	Assignment (%)
Mother alone			
Strict	95.0	0.00	20 (21)
Relaxed	80.0	0.00	36 (75)
Unassigned	-	-	12 (25)
Father alone			
Strict	95.0	0.00	8 (17)
Relaxed	80.0	0.00	36 (75)
Unassigned	-	-	12 (25)
Parent pair			
Strict	95.0	3.77	48 (100)
Relaxed	80.0	0.00	48 (100)
Unassigned	-	-	0 (0)

## Data Availability

The data sets generated and/or analyzed in the present study are available from the corresponding author upon reasonable request.

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
