# Peer review of "Developing a Microsatellite Polymerase Chain Reaction System for Small Yellow Croaker (Larimichthys polyactis) and Its Application in Parentage Assignment"

_biology, 2024, doi:10.3390/biology13090710_

Round 1
Reviewer 1 Report
Comments and Suggestions for Authors
In the manuscript, “Developing a microsatellites PCR system for small yellow croaker (Larimichthys polyactis) and their application in parentage assignment”. The authors identified 9 highly polymorphic SSR markers and designed a multiplex PCR system for the simultaneous amplification of these markers. These SSR markers were verified and confirmed for its application in parentage assignment in the small yellow croaker. The results were solid and meaningful for the selective breeding program of the small yellow croaker in future. I have some comments as following,
1. The information about the artificial insemination in table 1 is not very clear, you used 3 male and 3 female parents, how did you do the artificial insemination? You used 3 male to mate with 3 female or 1 male to mate with 3 female but with three replicates?
2. The authors identified 96 SSRs which was evenly distributed throughout the genome, but only 9 SSRs were involved in the multiplex PCR system, how about other SSRs?
3. As the authors have assembled the genome of the small yellow croaker, then why the authors chose to use microsatellite other than SNPs markers to do the parental assignment analysis.
4. It would be better to provide a table to include the number of alleles, HWE, null allele etc for the three mating groups.
Author Response
We would like to express our sincere gratitude for your invaluable assistance in reviewing and revising our submitted paper. We have carefully incorporated all of your suggestions and believe that the revised paper meets your expectations. Thank you once again for your support.
Q1. The information about the artificial insemination in table 1 is not very clear, you used 3 male and 3 female parents, how did you do the artificial insemination? You used 3 male to mate with 3 female or 1 male to mate with 3 female but with three replicates?
→ We have clarified the information on artificial insemination, as you recommended in line 93-98.
“A reference population comprising three families of small yellow croaker, each produced by artificial insemination from a mating pair (one male and one female), was used for marker validation. The final paternity test verification was conducted using a reference population generated through group mating”
Q2. The authors identified 96 SSRs which was evenly distributed throughout the genome, but only 9 SSRs were involved in the multiplex PCR system, how about other SSRs?
→ Out of the initial 96 candidate markers, a significant number were eliminated due to amplification failure, lack of polymorphism, or genotyping failure. Ultimately, only 9 markers were used in the analysis. These 9 markers were selected because they demonstrated successful amplification, sufficient polymorphism, and reliable genotyping results. Additionally, the consideration for multiplex PCR also played a role in the final selection of these 9 markers.
This content has been included in section 3.2. Microsatellite selection of the results.
Q3. As the authors have assembled the genome of the small yellow croaker, then why the authors chose to use microsatellite other than SNPs markers to do the parental assignment analysis.
→ Recently, research on the development of SNP markers for parentage verification has been conducted on various aquatic organisms as you mentioned. We have also submitted a paper on SNP markers and analysis methods for parentage verification of abalone (Haliotis discus hannai) to this journal, which is currently under review (biology-3169722).
When developing SNP markers for parentage verification, additional genome analysis from multiple individuals is required after genome sequencing. Generally, genome sequences are obtained through resequencing, and after mapping, the frequency of the nucleotide sequences is calculated to produce a SNP array containing a large number of SNPs. Genotype information is produced from parent-offspring samples using the SNP array, and markers with genetic linkage are selected to construct a set of SNPs for parentage verification.
Currently, parentage verification using SNPs is difficult at the aquaculture stage. It is considered more efficient to apply this when genomic selection is performed after reference group farming is stabilized.
Our team developed and applied SSR markers for this reason.
We hope this explanation is sufficient.
Q4. It would be better to provide a table to include the number of alleles, HWE, null allele etc for the three mating groups.
→ We have added a table 4 that includes information on the number of alleles, PIC, and null alleles as you recommended. In the development of parentage assignment markers, information of PIC is considered to be more important, and this content has been revised in the Results section in line 212-221.
Reviewer 2 Report
Comments and Suggestions for Authors
Current manuscript will be valuable for providing the useful information regarding to sustainable development in Aquaculture practices of yellow croaker. Overall your manuscript was satisfactory, however, please consider some suggestions to improve this article more as follows
Abstract and introduction were satisfied
Methodology section please add the refrences or citations of article from which you have adopted this methods.
Results satisfactory
Discussion portion required to improve more
Please add the details of other cited articles information regarding to the aquaculture of other fish species in table formate, this would attract the most readers for understanding more about your results compared with other published work.
Conclusions was written two times at the end of discussion
Wish you best of luck
Regards
Reviewer
Author Response
We would like to express our sincere gratitude for your invaluable assistance in reviewing and revising our submitted paper. We have carefully incorporated all of your suggestions and believe that the revised paper meets your expectations. Thank you once again for your support.
Q1. Methodology section please add the references or citations of article from which you have adopted this methods.
→ We have added references for the materials used in the methods section as you recommended. This includes DNA extraction [26] in line 86, genome assembly [27] in line 118, microsatellite discovery [29] in line 124, and statistical analysis [31] in line 168.
Q2. Please add the details of other cited articles information regarding to the aquaculture of other fish species in table format, this would attract the most readers for understanding more about your results compared with other published work.
→ We have incorporated the research findings on parentage analysis conducted in other species, as you recommended. However, we did not use a table and instead presented the information in sentence format in line 272-279. We appreciate your understanding.
Q3. Conclusions was written two times at the end of discussion.
→ We have revised the discussion section to avoid any duplication with the conclusion as you recommended.
Reviewer 3 Report
Comments and Suggestions for Authors
The paper ‘Developing a microsatellites PCR system for small yellow croaker (Larimichthys polyactis) and their application in parentage assignment’ by Eun Soo Noh with colleagues aims to develop a microsatellite marker-based multiplex PCR system for accurate parentage assignment for the small yellow croaker (Larimichthys polyactis), a species which is of significant economic value in East Asia. The main contribution to the field of science of this study is the development of a highly accurate system using genetic markers selected from the assembled genome from fish lines directly used in fisheries, which represents a valuable tool for breeding as well as sustainable aquaculture development. The study may be considered for publication in the Biology journal after addressing some of the weaknesses of the study and clarifying some of the issues. Specifically:
- The paper lacks references to software for genome assembly/annotation and searching for microsatellite markers;
- You indicated that you used COI for genetic identification of species. However, there is no mention of this in the results;
- There is no clear control group (e.g. non-mating individuals from natural conditions or from another farm) - which is another methodological drawback that would be good to address to ensure the reliability of the results obtained or to discuss in the paper for general benefit;
- While the paper mentions recent studies, it would be good to expand the discussion, especially with respect to alternative methods of parentage analysis (e.g. SNP-based systems) that could broaden the context of the study;
- Table 1: it is not clear why the number of individuals in the Group mating scheme is so much higher.
Author Response
We would like to express our sincere gratitude for your invaluable assistance in reviewing and revising our submitted paper. We have carefully incorporated all of your suggestions and believe that the revised paper meets your expectations. Thank you once again for your support.
Q1. The paper lacks references to software for genome assembly/annotation and searching for microsatellite markers.
→ We have added references for the materials used in the methods section as you recommended. This includes DNA extraction [26] in line 86, genome assembly [27] in line 118, microsatellite discovery [29] in line 124, and statistical analysis [31] in line 168.
Q2. You indicated that you used COI for genetic identification of species. However, there is no mention of this in the results.
→ We have added the analysis results of the mitochondrial COI gene in the Results section as you recommended in line 102-104.
Q3. There is no clear control group (e.g. non-mating individuals from natural conditions or from another farm) - which is another methodological drawback that would be good to address to ensure the reliability of the results obtained or to discuss in the paper for general benefit.
→ We conducted a reanalysis by incorporating research results on a separate control group in the paternity verification as you recommended. The relevant details have been added to Figure 3, Table 4-5, and Result 3.3. in line 207-208, and 224-230.
Q4. While the paper mentions recent studies, it would be good to expand the discussion, especially with respect to alternative methods of parentage analysis (e.g. SNP-based systems) that could broaden the context of the study.
→ We have added a section to the conclusion regarding the expansion of research into parentage assignment using SNPs as you recommended in line 317-325.
Q5. Table 1: it is not clear why the number of individuals in the Group mating scheme is so much higher.
→ The reason for the large number of individuals in the group mating is for the final verification of the analysis markers. This is because a more complex offspring generation than the existing 1:1 artificial insemination is produced, and a large number of analyses are required to ensure accuracy. This also includes sibling or half-sibling relationships, which is why a large number were analyzed. The relevant details have been added to the results section in line 247-254.